# Sharpe Ratio-Guided Active Learning for Preference Optimization in RLHF

**Syrine Belakaria**[*], **Joshua Kazdan**[*], **Charles Marx**[*], **Chris Cundy**[*], **Willie Neiswanger**[‡],
**Sanmi Koyejo**[*], **Barbara E. Engelhardt**[*+], **Stefano Ermon**[*]

Stanford University[*]
Gladstone Institutes[+]
University of Southern California[‡]

{syrineb, jkazdan, ctmarx, cundy, sanmi, bengelhardt, ermon}@stanford.edu
{neiswang}@usc.edu

## Abstract

Reinforcement learning from human feedback (RLHF) has become a cornerstone of the training and alignment pipeline for large language models (LLMs). Recent advances, such as direct preference optimization (DPO), have simplified the preference learning step. However, collecting preference data remains a challenging and costly process, often requiring expert annotation. This cost can be mitigated by carefully selecting the data points presented for annotation. In this work, we propose an active learning approach to efficiently select prompt and preference pairs using a risk assessment strategy based on the Sharpe Ratio. To address the challenge of unknown preferences prior to annotation, our method evaluates the gradients of all potential preference annotations to assess their impact on model updates. These gradient-based evaluations enable risk assessment of data points regardless of the annotation outcome. By leveraging the DPO loss derivations, we derive a *closed-form expression* for computing these Sharpe ratios on a per-tuple basis, ensuring our approach remains both *tractable* and *computationally efficient*. We also introduce two variants of our method, each making different assumptions about prior information. Experimental results demonstrate that our method outperforms the baseline by up to 5% in win rates against the chosen completion with limited human preference data across several language models and real-world datasets.

## 1 Introduction

Reinforcement Learning from Human Feedback (RLHF) constitutes the final step of training for modern large language models (LLMs) (Christiano et al., 2017a). RLHF ensures that language models align with human preferences in many aspects, including response length (Singhal et al., 2024), helpfulness (Li et al., 2024), and lack of harmfulness. RLHF can be used to align models according to any criterion of choice from the user and has been extended beyond language to vision (Yang et al., 2024; Wallace et al., 2024) and scientific models (Gu et al., 2025). However, unlike pretraining data, which can be scraped in large quantities from sources such as books, archives, and the internet without requiring annotation, RLHF data is costly to gather, as it necessitates expert labeling depending on the specific domain (Bai et al., 2022c; Lee et al.).

RLHF data is generally structured as tuples consisting of a single prompt and multiple candidate responses. In an ideal setup, one response within each tuple is labeled as *preferred*, while the remaining responses are marked as *rejected*. Due to the potentially large volume of such tuples, however, labeling them all is prohibitively expensive and impractical. As a result, only a limited subset of these data points can typically be presented to expert annotators. Established RLHF datasets, for instance, often include only a few tens of

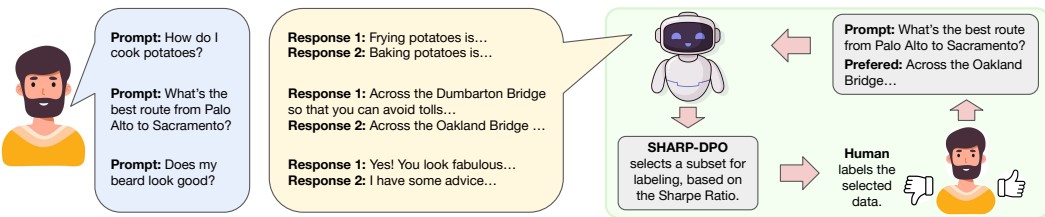

Figure 1: Workflow for pool-based active learning in DPO. First, a user asks the LLM questions. The LLM generates two candidate answers to each question. A subset of the question-responses tuples are chosen for labeling by the user. Then, the model is updated using the collected human preferences.

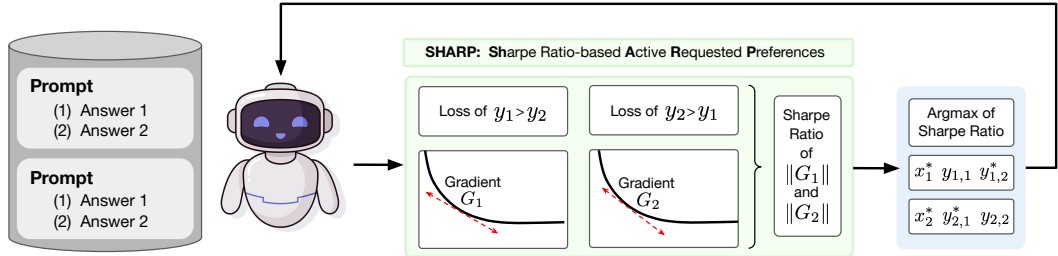

Figure 2: An illustration of the steps of active learning for RLHF using Sharpe Ratio selection criteria.

thousands of these expert-labeled preference pairs, despite the much larger volume of unlabeled data available (Bai et al., 2022a; Ethayarajh et al., 2022).

The high cost of producing RLHF fine-tuning data leads to investigating more efficient data collection strategies. Models generate millions of responses to human prompts each day; among these, which prompts—if labeled with preference pairs—would provide the greatest benefit during additional RLHF training? Identifying the prompt-responses triplets that yield the highest impact on training could substantially reduce both the time and monetary costs associated with human annotation. This question falls under the broader umbrella of *Active Learning*(AL), which aims to determine the most informative samples for model training. (Ren et al., 2021).

Active learning algorithms have demonstrated success for both general statistical models (Castro et al., 2005; Tong & Koller, 2000) and deep learning models under supervised learning (Ren et al., 2021). However, relatively few approaches focus on applying AL to RLHF for LLMs. In recent work, Muldrew et al. (2024) selected prompt-responses triplets by prioritizing higher reward gaps, while Mehta et al. (2023) used uncertainty metrics to target data where the model appeared less confident. Both of these methods implicitly rely on predicting which response will be preferred, incorporating that assumption directly into the selection process. In contrast, our approach accounts for all potential preference outcomes, enabling the assessment of data points regardless of which response is chosen by the expert.

More recently, direct preference optimization (DPO) was proposed as an alternative to the traditional RLHF pipeline that simplifies the process of learning from preference-labeled data (Rafailov et al., 2023a). In this work, we present a novel active learning technique that targets an effective selection of data for DPO. We propose to leverage information about the magnitude of gradient update as a selection criterion. Before gathering human preferences about which of two responses is favored, we note that the gradient update will assume one of two forms, depending on which response is set as chosen. As each of these responses is equally likely to be preferred, the resulting gradient update can be seen as a random variable that will settle to one of two values. Rather than relying solely on the expectation or variance of this random variable, we draw inspiration from statistical finance and adopt the Sharpe ratio to characterize and compare the potential updates(Sharpe, 1998). The Sharpe ratio naturally balances the expected improvement (mean) against the uncertainty (standard deviation), making it well-suited to pinpoint samples that promise substantial gains while

managing risk. Accordingly, we select prompt–responses triplets that yield the highest Sharpe ratios, focusing on cases with the greatest potential for informativeness.

Importantly, we propose a derivation that allows us to obtain a closed-form expression for per-tuple Sharpe ratios, circumventing the need for the memory and computationally intensive multiple backpropagations and keeping our method tractable and efficient. We further introduce two variants of our approach. The first, SHARP (**SH**arpe Ratio-based **A**ctive **R**equested **P**references), assumes all possible annotations are equally likely. The second, W-SHARP, incorporates the implicit reward model as a prior, producing a weighted version of SHARP that accounts for varying annotation likelihoods.

By applying our procedures, we achieve up to 5% improvement in win rate over the benchmark dataset's preferred completions, even with a highly constrained data budget, less than 18% of available training tuples in the HH (Bai et al., 2022b) and SHP (Ethayarajh et al., 2022) datasets. We demonstrate the effectiveness of our algorithm across different model scales—specifically Llama-3-8B and Pythia-2.8B—using two state-of-the-art benchmarks: the Helpful-Harmless (HH) dataset (Bai et al., 2022b) and the Stanford Human Preferences (SHP) dataset (Ethayarajh et al., 2022).

To summarize our contributions:

- Drawing inspiration from statistical finance, we introduce a risk assessment approach for active learning in RLHF/DPO. Our method uses the Sharpe ratio of gradient magnitudes to determine which data points are most valuable for labeling.

- We propose two instantiations of our proposed method. The first assumes that each response is equally likely to be chosen as preferred, while the second uses a prior derived from an implicit reward model to weigh the likelihood of each response.

- Leveraging the DPO loss function, we derive fast and memory-efficient closed-form expressions of our acquisition functions.

- We demonstrate improvements in win rates on popular RLHF datasets using three different LLMs of varying sizes.

## 2  Background

In this section, we review the details of RLHF and direct preference optimization (DPO). Reinforcement Learning from Human Feedback (RLHF) has emerged as a key approach for aligning language models with human preferences. Originally popularized by works such as Christiano et al. (2017b) and Stiennon et al. (2020), the standard RLHF pipeline begins with a supervised fine-tuning (SFT) phase using high-quality data, followed by training a reward model on preference-labeled examples. In the final phase, the policy is further refined through reinforcement learning, where the reward model, reflecting human feedback, serves as a learned utility function guiding the policy updates via algorithms like Proximal Policy Optimization (PPO) (Schulman et al., 2017; Shao et al., 2024).

A major drawback of traditional RLHF was the need to train a reward function, which increases the computational complexity of the alignment step due to the overhead of a separate model. Additionally, reward models are often large, unstable, and might overfit to the preference data (Skalse et al., 2022; Yan et al., 2024; Chaudhari et al., 2024). To obviate the need to train a reward function, Rafailov et al. (2023b) developed direct preference optimization (DPO), an adaptation of the Bradley-Terry model (Bradley & Terry, 1952) that converts the RLHF pipeline into a preference classification problem and uses the language model and the reference model to form an implicit reward model. Specifically, let $x$ be a prompt and $y$ be a response to this prompt. Denoting the policy model by $\varphi_\theta$ and the reference model by $\varphi_{\text{ref}}$, The RLHF optimization problem is expressed as:

$$\max_{\varphi_\theta} \mathbb{E}_{x \sim \mathcal{D}, y \sim \varphi_\theta(y|x)} \left[ r_\phi(x, y) \right] - \beta \mathbb{D}_{\text{KL}} \left[ \varphi_\theta(y \mid x) \mid\mid \varphi_{\text{ref}}(y \mid x) \right] \tag{1}$$

The optimal solution to the KL-constrained reward maximization objective leads to an expression of the reward model as:

$$r(x, y) = \beta \log \frac{\varphi_\theta(y|x)}{\varphi_{\text{ref}}(y|x)} + \beta \log Z(x). \tag{2}$$

In this equation, $\beta$ is a hyper-parameter that controls the deviation of the policy from the reference policy, and $Z(x)$ is the partition function that depends only on $x$. Let $r^*$ be the ground-truth reward, and $\varphi^*$ be the optimal policy. Under the Bradley-Terry model (Bradley & Terry, 1952), the probability that one response is preferred over another is:

$$p^*(y_1 \succ y_2|x) = \sigma(r^*(x, y_1) - r^*(x|y_2)). \tag{3}$$

Substituting in Equation equation 2, the preference probabilities under Bradley-Terry model can be expressed as a function of the optimal RLHF policy $\varphi^*$ as follows:

$$p^*(y_1 \succ y_2|x) = \frac{1}{1 + \exp\left(\beta \log \frac{\varphi^*(y_2|x)}{\varphi_{\text{ref}}(y_2|x)} - \beta \log \frac{\varphi^*(y_1|x)}{\varphi_{\text{ref}}(y_1|x)}\right)}. \tag{4}$$

Since we can express the probability of human preference data in terms of the optimal policy rather than a separate reward model, we can construct a maximum likelihood objective for a parameterized policy $\varphi_\theta$ in terms of the chosen $y_w$ and rejected $y_\ell$ rewards. This produces a preference classification loss function:

$$\mathcal{L}_{\text{DPO}}(x, y_w, y_l) = -\log \sigma\left(\beta \log \frac{\varphi_\theta(y_w \mid x)}{\varphi_{\text{ref}}(y_w \mid x)} - \beta \log \frac{\varphi_\theta(y_l \mid x)}{\varphi_{\text{ref}}(y_l \mid x)}\right). \tag{5}$$

While training using the DPO objective, one simultaneously trains the language model and an implicit reward model. This saves substantial time and computation by removing the need to train a separate reward model. In this work, we develop an active learning method for RLHF. Although we experimentally focus on DPO due to its lower computational overhead, our method also applies to RLHF.

## 3 Related Work

Some estimates suggest that over 80% of engineering efforts in machine learning concern data preparation and labeling (Fredriksson et al., 2020). Active learning (AL), also referred to as optimal experimental design (Olsson, 2009), aims to achieve strong model performance with fewer training samples (Alizadeh et al., 2021). The most common use case for active learning occurs when there is a large pool of unlabeled data, and the scientist training a machine learning model must choose which of these data points should be labeled, subject to a labeling budget. In AL, an *acquisition function* applied to the unlabeled data points is used to perform this selection. AL techniques have been applied across various machine learning domains such as support vector machines (SVM) (Tong & Koller, 2001), image classification (Gal et al., 2017), and other areas (Settles, 2009). Recent efforts in deep active learning (DAL) have focused on text classification (Tuia et al., 2011), image analysis (Wang et al., 2023), and NLP (Hadian & Sameti, 2014). Many active learning methods are based on the principle of uncertainty (Tong & Koller, 2001), wherein the algorithm prioritizes labeling data points that the model is most uncertain about. Other active learning methods emphasize the importance of diversity and exploration when choosing different types of examples to label (Doucet et al., 2024).

Across domains, AL is a notoriously difficult problem (Hanneke & Yang, 2010; Castro & Nowak, 2007). Active learning is especially challenging for RLHF in large-scale models that lack convexity guarantees or bounded noise. Currently, few works tackle the design of acquisition functions in this context. Recently, Mehta et al. (2023) and Ji et al. (2024) formulated active learning for RLHF and DPO as an offline contextual dueling bandit problem. Mehta et al. (2023) proposed an uncertainty-based approach, measuring variance in predicted logits under dropout, while Ji et al. (2024) introduced an algorithm with theoretical

guarantees on regret and query complexity. In parallel, Muldrew et al. (2024) explored active learning in DPO by first selecting a sub-batch of prompts with high predictive entropy, then further filtering based on large reward gaps, interpreted as lower uncertainty in the DPO model. Although these methods employ different exploration or exploitation strategies, they often require a prior assumption about which response is preferred, computing acquisition scores under that assumption. Ideally, an active learning approach should consider *all* possible preference outcomes without relying on a predefined guess. Our method fulfills this criterion, offering a first attempt at a risk-based perspective that balances exploration and exploitation more comprehensively.

Beyond dueling bandit frameworks, Zhang et al. (2024) introduced a bilevel optimization approach for DPO that favors potentially high-reward responses. Xiong et al. (2024) proposed an online exploration method and a rejection sampling strategy for offline settings, formulated as a reverse-KL-regularized contextual bandit.

## 4 Problem Setting

Consider a practitioner who wishes to fine-tune a large language model (LLM) via reinforcement learning from human feedback (RLHF) in a specific domain. The practitioner has access to a large pool of *unlabeled data*,

$$\mathcal{D} = \left\{ (x_i, y_{i1}, y_{i2}) \right\}_{i=1}^n$$

where $n$ is large, and each entry consists of a *prompt* $x_i$ along with two *candidate responses* $y_{i1}$ and $y_{i2}$. Owing to the high cost and impracticality of labeling every entry in $\mathcal{D}$, only a small subset $\mathcal{D}_L \subseteq \mathcal{D}$ can be annotated with *expert preferences* (i.e., which of $y_{i1}$ or $y_{i2}$ is preferred).

Once the practitioner obtains $b$ *labeled triplets* from $\mathcal{D}_L$, a *direct preference optimization (DPO)* update is performed on the LLM. The model is then used to query a new batch of unlabeled data for expert feedback, and this iterative process continues until the labeling budget is exhausted. The key challenge is to *select the most informative triplets* for labeling to maximize the final performance of the RLHF-fine-tuned model under strict budget constraints.

To closely mirror practical scenarios of collecting and deploying preference data, we require a criterion that identifies the most valuable prompts for human annotation. In our experimental setup, we model this situation as follows:

1. For each prompt and response pair in a large batch of size $b \times p$, evaluate a designed selection criterion, where $p$ is a user-defined fraction indicating the annotation budget. We use this strategy as a practical search procedure.

2. Rank all triplets based on the selection criterion and select the top $b$ to label.

3. Using the labeled preference pairs and perform a single DPO update.

## 5 Sharpe Ratio for Active Preference Learning

### 5.1 Method Description

We propose a novel method to efficiently collect human preference data in an online setting. Our strategy maximizes the gradient magnitude derived from the DPO objective on the selected data, thereby using information about model parameters when deciding which samples will have the greatest training impact.

A key challenge arises because we cannot compute the DPO gradient without knowing which response is actually preferred. However, we do know that, for each prompt $x$ with candidate responses $y_1$ and $y_2$, the gradient will assume exactly one of two possible forms: one if $y_1$ is preferred, and another if $y_2$ is preferred. Let $\varphi_{\text{ref}}$ denote the reference model. Depending on which response is ultimately chosen, the DPO update takes one of the

following two forms:

$$G_1 = \nabla_\theta \mathcal{L}_{\text{DPO}}(x, y_1, y_2) = -\nabla_\theta \log \sigma \Big( \beta \log \frac{\varphi_\theta(y_1|x)}{\varphi_{\text{ref}}(y_1|x)} - \beta \log \frac{\varphi_\theta(y_2|x)}{\varphi_{\text{ref}}(y_2|x)} \Big)$$

$$= -\beta \sigma(\hat{r}_\theta(x, y_2) - \hat{r}_\theta(x, y_1)) \times \big[ \nabla_\theta \log \varphi_\theta(y_1|x) - \nabla_\theta \log \varphi_\theta(y_2|x) \big]$$

$$G_2 = \nabla_\theta \mathcal{L}_{\text{DPO}}(x, y_2, y_1) = -\nabla_\theta \log \sigma \Big( \beta \log \frac{\varphi_\theta(y_2|x)}{\varphi_{\text{ref}}(y_2|x)} - \beta \log \frac{\varphi_\theta(y_1|x)}{\varphi_{\text{ref}}(y_1|x)} \Big)$$

$$= -\beta \sigma(\hat{r}_\theta(x, y_1) - \hat{r}_\theta(x, y_2)) \times \big[ \nabla_\theta \log \varphi_\theta(y_2|x) - \nabla_\theta \log \varphi_\theta(y_1|x) \big].$$

Let $G$ be the random variable defined by the magnitude of the gradient update that is obtained by soliciting human feedback for the $(x, y_1, y_2)$ triplet. Let $p_1 = p(y_1 \succ y_2|x)$ be the probability that $y_1$ is preferred to $y_2$ and $p_2 = p(y_2 \succ y_1|x)$ be the probability that $y_2$ is preferred to $y_1$.

The expectation of $G$ is defined as:

$$\mathbb{E}[G] = p_1\|G_1\| + p_2\|G_2\|. \tag{6}$$

The variance of G is defined as:

$$\sigma^2(G) = p_1(\|G_1\| - \mathbb{E}[G])^2 + p_2(\|G_2\| - \mathbb{E}[G])^2. \tag{7}$$

The expectation alone is not a good decision metric when selecting which responses should be labeled for several reasons. First, suppose that one response is gibberish, and the other is sensible. The gradient in which the gibberish response is the preferred response would likely be large, and therefore, the expectation would be high. However, selecting a tuple where one of the responses is gibberish will not lead to an informative update to the model. Thus, we need some way to account for the variance of $G$. To do this, we use a tool from statistical finance: the Sharpe ratio. The Sharpe ratio (Sharpe, 1966), invented by William Sharpe in the 1960s, evaluates not just the expected value of an investment portfolio but also the risk. For example, one would likely eschew investment opportunities that could result in losing one's entire life savings, even if these investment opportunities had a high expected value. We apply the same logic when selecting which preference pairs to label. We want to maximize the expected magnitude of our gradient updates but reduce the risk of getting a small gradient update if a certain response is preferred. By choosing to label the preference pairs that yield the highest Sharpe ratio, we accomplish this goal. Because we drew inspiration for our method of active learning from the Sharpe ratio metric, we name our method **SH**arpe Ratio-based **A**ctive **R**equested **P**references, or **SHARP** for short. The Sharp ratio of a triplet $(x, y_1, y_2)$ is defined as:

$$SR(G) = \frac{\mathbb{E}[G]}{\sigma(G)} \tag{8}$$

In our active learning setting, we select triplets that yield the highest Sharpe ratio. We define an acquisition function for the current policy $\varphi_\theta$ as:

$$\alpha_{\varphi_\theta}(x, y_1, y_2) = SR(G). \tag{9}$$

**SHARP: No Prior Acquisition Function**: Before querying the expert labeling of the preference, we might have no prior assumption about which response might be preferred to the other. In this case, we can assume that $y_1$ and $y_2$ are equally likely to be the better response, and therefore $p_1 = p_2 = 0.5$. We consider this the no prior version of our method, and we refer to it as SHARP.

**W-SHARP: Prior-based Acquisition Function**: The RLHF/DPO pipeline usually initializes the policy $\varphi_\theta$ to the SFT policy previously finetuned on data related to the same domain or topic of interest. This model can provide us with a prior for the preference probabilities $p_1$ and $p_2$. For instance, in the DPO setting, we can derive an implicit reward model from $\varphi_\theta$ and $\varphi_{\text{ref}}$, $r_\theta(x, y) = \beta \log \frac{\varphi_\theta(y|x)}{\varphi_{\text{ref}}(y|x)}$ and then set the probabilities $p_1$ and $p_2$ based on Equation 4 during the active learning iterations. We refer to this version of our method as weighted SHARP (W-SHARP).

## 5.2 Efficient Execution of SHARP with DPO

In practice, computing the Sharpe ratio would require computing the gradient for each element in the dataset twice and consequently backpropagating through the LLM $2 \times B$ times for each batch of size $B$ instead of a single batch-wise backpropagation. This procedure is computationally expensive in terms of both time and memory. To overcome this bottleneck, we use the closed-form expression of the gradient of the DPO loss function to simplify the final expression of the SHARP acquisition functions. Given the final expression of the gradient of the DPO loss, we can express $G_2$ as a function of $G_1$:

$$
\begin{aligned}
G_2 &= -\beta \sigma(\hat{r}_\theta(x, y_1) - \hat{r}_\theta(x, y_2)) \times \left[ \nabla_\theta \log \varphi_\theta(y_2|x) - \nabla_\theta \log \varphi_\theta(y_1|x) \right] \\
&= -\beta \left[ \sigma(\hat{r}_\theta(x, y_2) - \hat{r}_\theta(x, y_1)) - 1 \right] \times \left[ \nabla_\theta \log \varphi_\theta(y_1|x) - \nabla_\theta \log \varphi_\theta(y_2|x) \right] \\
&= G_1 \left[ 1 - \frac{1}{\sigma(\hat{r}_\theta(x, y_2) - \hat{r}_\theta(x, y_1))} \right].
\end{aligned}
$$

Consequently, we have:

$$
\|G_2\| = \|G_1\| \cdot \|\gamma\|, \tag{10}
$$

with $\|\gamma\| = \|1 - \frac{1}{\sigma(\hat{r}_\theta(x, y_2) - \hat{r}_\theta(x, y_1))}\|$.

Combining Equations 8, Equation 6, and Equation 7, we get an expression of the Sharpe ratio as follows:

$$
SR(G) = \frac{p_1 \|G_1\| + p_2 \|G_2\|}{\sqrt{p_1(\|G_1\| - \mathbb{E}[G])^2 + p_2(\|G_2\| - \mathbb{E}[G])^2}}.
$$

By substituting the expression of $\|G_2\|$ from Equation 10, we obtain the final form of the Sharpe ratio, in which the gradient terms $\|G_1\|$ cancel out in both the numerator and the denominator.

$$
SR(G) = \frac{\|G_1\|(p_1 + p_2\|\gamma\|)}{\sqrt{p_1(\|G_1\| - \|G_1\|(p_1 + p_2\|\gamma\|))^2 + p_2(\|G_1\| \cdot \|\gamma\| - \|G_1\|(p_1 + p_2\|\gamma\|))^2}} \tag{11}
$$

$$
= \frac{(p_1 + p_2\|\gamma\|)}{\sqrt{p_1(1 - (p_1 + p_2\|\gamma\|))^2 + p_2(\|\gamma\| - (p_1 + p_2\|\gamma\|))^2}}. \tag{12}
$$

In the case of W-SHARP, we substitute the probabilities $p_1$ and $p_2$ by the preference probabilities obtained by combining the implicit reward model and the Bradley-Terry preference model (Bradley & Terry, 1952):

$$
\alpha_{\varphi_\theta}^{W-SHARP}(x, y_1, y_2) = SR(G), \tag{13}
$$

with $SR(G)$ defined in Equation 12. In the case of SHARP, where we assume that we do not have any prior about the preference probabilities, we have $p_1 = p_2 = \frac{1}{2}$. The acquisition function expression can be further simplified as follows:

$$
\alpha_{\varphi_\theta}^{SHARP}(x, y_1, y_2) = \frac{\frac{1}{2}(1 + \|\gamma\|)}{\sqrt{\frac{1}{2}(1 - \frac{1}{2}(1 + \|\gamma\|))^2 + \frac{1}{2}(\|\gamma\| - \frac{1}{2}(1 + \|\gamma\|))^2}}. \tag{14}
$$

By simplifying Equation 14, we obtain the final expression:

$$
\alpha_{\varphi_\theta}^{SHARP}(x, y_1, y_2) = \frac{1 + \|\gamma\|}{|1 - \|\gamma\||}. \tag{15}
$$

By leveraging the gradient expression of the DPO loss and the relationship between swapped-preference gradients and the Sharpe ratio, our derivation provides a **closed-form

formula** for per-tuple Sharpe ratios. This circumvents the need for multiple backpropagations, significantly reducing both memory and computation costs and keeping the method tractable. Crucially, without this derivation, although the approach might still be conceptually valid and useful, it would be prohibitively impractical in real-world applications.

We execute SHARP and W-SHARP on each batch of incoming unlabeled prompt-responses triplets to select a sub-batch for human labeling. SHARP proceeds as in Algorithm 1.

---

**Algorithm 1** SHARP Data Selection Algorithm

---

**Inputs:** policy $\varphi_\theta$, reference policy $\varphi_{\text{ref}}$, exploration parameter $\beta$, batch size $b$, number of iterations $N$, a dataset $\mathcal{D} = \{(x_i, y_{i1}, y_{i2})\}_{i=1}^n$, the fraction $p$ of the batch that we can afford to label.
**Output:** A subset of the data $\mathcal{D}_L \subset \mathcal{D}$ of triplets of expert labeling with $|\mathcal{D}_L| = b \times N$, Updated $\varphi_\theta$.

1: **for** $t = 1, \ldots, N$ **do**
2:      Draw a large batch of triplets $B = \{(x_i, y_{i1}, y_{i2})_{i=1}^{(b.p)}\} \sim \mathcal{D}$.
3:      **for** $(x_i, y_{i1}, y_{i2}) \in B$ **do**
4:          If using SHARP method, compute $\alpha_{\varphi_\theta}^{SHARP}(x_i, y_{i1}, y_{i2})$
5:          If using W-SHARP method, compute $\alpha_{\varphi_\theta}^{W-SHARP}(x_i, y_{i1}, y_{i2})$
6:      **end for**
7:      Let $B_L$ be the top-$b$ elements of $B$ by the value of the acquisition function $\alpha$.
8:      Request the preferences labels from the expert and add them to $\mathcal{D}_L$
9:      Update the policy $\varphi_\theta$ using a gradient step of the $\mathcal{L}_{\text{DPO}}$ using $B_L$
10: **end for**
11: **return** $\mathcal{D}_L$ and $\varphi_\theta$.

---

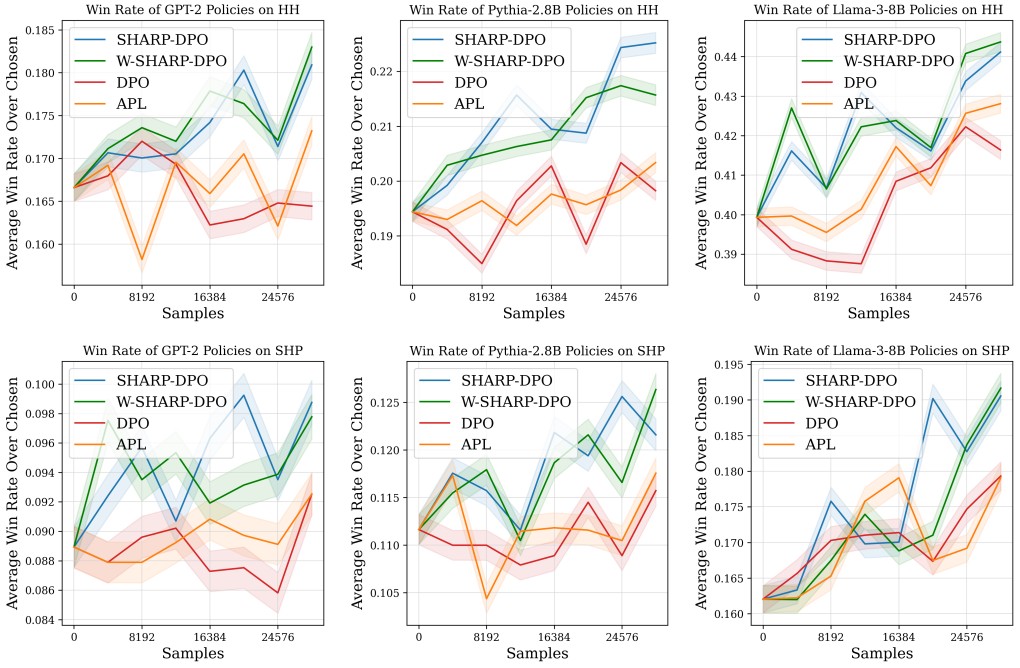

Figure 3: Comparison of W-SHARP-DPO and SHARP-DPO against DPO and APL across different models and datasets. The metric is the average win rate over chosen completions, computed with GPT-4o under swapped evaluation orders to reduce positional bias. Error bars indicate the standard error.

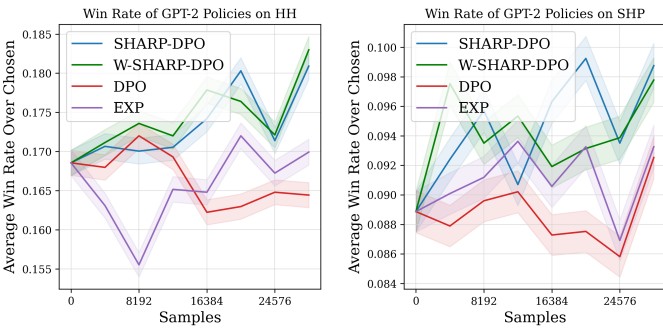

Figure 4: Comparison against the expectation over the gradient on the GPT-2 model with both datasets.

## 6 Experiments

In this section, we provide the details of our evaluation pipeline. Our main goal is to determine if we can achieve better or comparable performance as DPO while using a smaller amount of labeled data. The standard DPO uses a random selection from the dataset. To assess whether our approaches enhance data selection in DPO, we conduct experiments training large language models (LLMs) on two datasets applied to three different models with different ranges of sizes. We additionally provide comparison results against the APL baseline proposed by Muldrew et al. (2024). When comparing the approaches, we keep all parameters of the experiments identical except for the data selection method to isolate and verify its impact on performance. The code for our approach is publicly available github.com/belakaria/sharpe-ratio-active-llm-alignment-dpo.

**Datasets** We evaluate both methods on two public datasets: the Anthropic Helpful-Harmless (HH) dataset (Bai et al., 2022b) and the Stanford Human Preferences (SHP) dataset (Etha-yarajh et al., 2022).

*Anthropic Helpful-Harmless (HH)*: The HH dataset is designed to measure an AI assistant's ability to be both helpful and harmless. It contains two main types of examples: queries where the user's request is reasonable and the assistant should provide a helpful response, and queries where the user's request may be harmful or inappropriate, requiring the assistant to prioritize safety by giving a non-harmful response.

*Stanford Human Preferences (SHP)*: The SHP dataset consists of Reddit posts and corresponding human-generated comments spanning 18 different categories. This broad coverage provides diverse human writing styles and topics. SHP focuses on modeling general human preferences across a wide range of real-world conversations.

**LLMs**: We explore the impact of active learning by evaluating three models of varying size and capacity: GPT-2, Pythia-2.8-B, and Llama-3-8B. These models span a broad range of resource requirements and capabilities, allowing us to assess how active learning strategies perform under different constraints. We conduct six distinct experiments using the above datasets to provide a comprehensive analysis of each model's performance.

**Pipeline Setup**: In the DPO pipeline, we begin by splitting each dataset into training and test sets. During the Supervised Fine-Tuning (SFT) phase, we finetune each model on the training split, updating all parameters in each gradient step. In the subsequent DPO phase, to efficiently manage computational resources, we apply a quantized Low-Rank Adaptation (LoRA) of each LLM. This approach reduces memory footprint and speeds up experimentation without sacrificing the model's overall performance. We apply 4-bit quantization using a double quantization strategy under the NF4 scheme while computing in bfloat16. In addition, we use a LoRA configuration with rank 16, alpha 32, and a dropout rate of 0.05, tailored for causal language modeling tasks and omitting additional bias. We set the batch size of our training to 64 and set the fraction that would be labeled to $p = 6$.

We evaluate model performance using the winrate against the dataset's designated "chosen" completions. Formally, the winrate indicates the proportion of generated responses that are

deemed preferable to those labeled as chosen in the dataset. We recompute this metric after every 4,096 training samples to track performance trends over time.

To underscore the benefits of active learning under constrained resources, we limit the DPO phase to a total of 28,672 training points across all experiments. Additionally, we use GPT-4o as an evaluation oracle to compare each newly generated response against the dataset's designated chosen completions. To mitigate position bias, each pair of responses is evaluated twice with reversed ordering, and we report the average winrate across these two evaluations.

Both W-SHARP-DPO and SHARP-DPO consistently outperform the standard DPO baseline and the APL baseline (Figure 3). We attribute this improvement to our acquisition function $\alpha$, which takes the risk (i.e., all possible gradient outcomes) into account when selecting data points. Interestingly, W-SHARP-DPO and SHARP-DPO achieve similar performance, suggesting that incorporating the implicit reward model as a prior does not necessarily yield further gains in this setting. This could indicate that while using a prior might help in other contexts, it is not required for effective data selection and making no prior assumption could be beneficial for risk assessment.

The accuracy of the implicit reward model for experiments conducted on both datasets echos this result (Figure 5, Appendix). Although this metric is not our primary focus, the results indicate that both SHARP and W-SHARP tend to attain higher accuracy more quickly on the test data, suggesting that these methods guide the model toward more effective reward predictions.

**Ablation Study**: We additionally provide an ablation study to compare against the expected gradient (Figure 4). We conduct the experiments on the GPT-2 model using both datasets. We observe that the SHARP method performs better. Notably, we expect the performance gap to be even larger in noisy datasets that include response pairs where one option is unlikely or nonsensical. Due to the high memory and computational demands of extracting individual gradients for each data point, we were only able to run these experiments with the GPT-2 model. This computational challenge further underscores the strength of the SHARP approach, as its closed-form expression avoids the need for expensive per-sample gradient computations.

## 7 Summary, Future Directions, and Limitations

We present a novel active learning strategy for RLHF/DPO in large language models, designed to prioritize and label the most impactful data points under limited human annotation budgets. Central to our method is the use of a Sharpe ratio-based acquisition function to evaluate potential gradient updates. By selecting examples with the highest Sharpe ratios, we aim to target those most likely to produce substantial improvements in policy performance. Our empirical results suggest that this risk-aware selection can reduce annotation costs while enhancing the quality of the learned policy.

Our current approach focuses exclusively on high Sharpe ratio data, which may bias the distribution of selected examples. Although such selective sampling is typical in active learning scenarios, if a practical setting requires an unbiased estimate of the underlying data distribution, future methods could address potential deviations arising from risk-based sampling. Potentially, future methods could combine our Sharpe ratio-based approach with techniques like importance sampling or explicit expectation balancing to address such requirements. Moreover, our computational study was limited by relatively modest resources, restricting the scale of DPO training and the range of datasets evaluated. While our findings demonstrate the promise of a Sharpe ratio-based framework, additional investigation across larger tasks and more extensive experiments would establish its robustness and generalizability.

Although in this work we focused on Sharpe ratio, given its intuitive trade-off between the expected benefit (mean gradient magnitude) and the potential downside (variance), alternative risk-aware metrics such as the Sortino ratio could offer interesting inductive biases. Investigating the impact of such metrics is a promising direction for future work.

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

# Appendix

## A    Additional Results

In this section, we provide additional results reporting the accuracy of the implicit reward model on the test set. To provide informative results, we use exponential moving average smoothing.

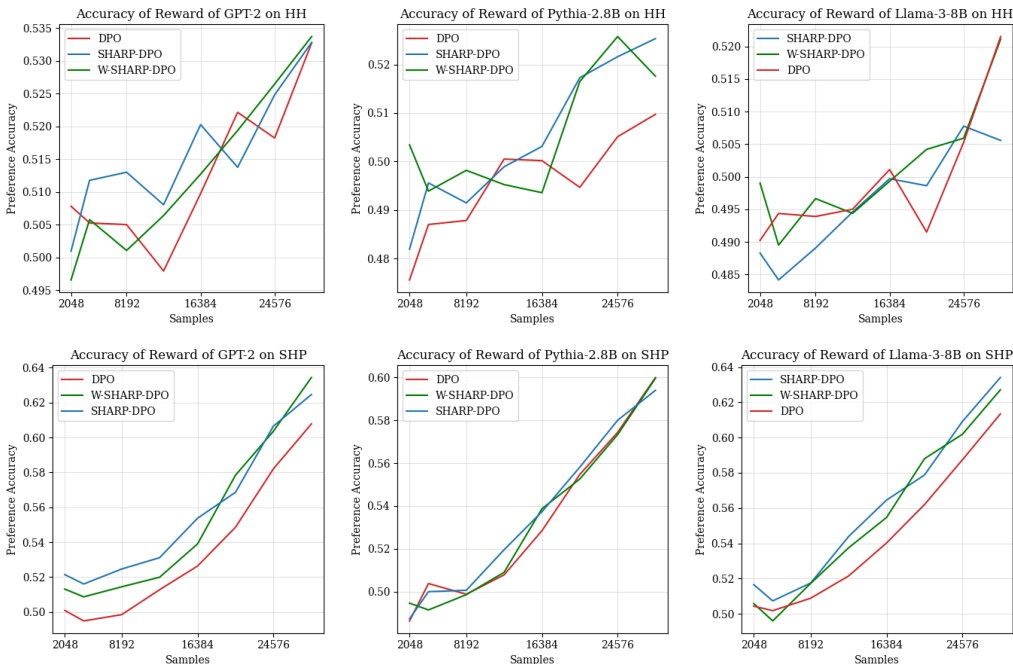

Figure 5: The accuracy of the implicit reward model for models.

