# OpenReview forum: "Sharpe Ratio-Guided Active Learning for Preference Optimization in RLHF"
_colmweb.org/COLM/2025/Conference — COLM 2025_

### Official Review · Reviewer_C2ud · 2025-05-10

**Rating:** 4
**Confidence:** 4
**Ethics Flag:** 1

**Summary:**

The paper proposes a novel active learning method for RLHF/DPO in LLMs, leveraging the Sharpe ratio—a financial risk-assessment metric—to prioritize data points that balance expected gradient update magnitudes and their variance. Two variants, SHARP (no prior) and W-SHARP (with reward model prior), are introduced. The authors derive closed-form expressions for efficient computation, avoiding multiple backpropagations. Experiments on the HH and SHP datasets demonstrate improved win rates over baselines with constrained data budgets (18% of training data), validated across models like Llama-3-8B and Pythia-2.8B.

**Reasons To Accept:**

Innovative Methodology: The application of the Sharpe ratio to active learning is novel, effectively addressing the exploration-exploitation trade-off in RLHF data selection.

Computational Efficiency: The closed-form gradient derivation reduces memory and computation costs, making the method scalable for large models.

**Reasons To Reject:**

Theoretical Limitations: The choice of the Sharpe ratio lacks rigorous theoretical justification compared to alternative risk-aware metrics (e.g., Sortino ratio). Why the authors choose sharpe ratio as the foundation metric? What happened if we choose other ones?

Narrow Evaluation: Experiments are limited to two datasets and lack diversity in tasks (e.g., harmlessness vs. general preferences). Broader validation is needed.

Evaluation Concerns: Reliance on GPT-4o as an oracle evaluator introduces potential positional bias, which is only partially mitigated by order swapping.

Behind Reason: I don't think that the gradient of loss function measures the importance or quality of the samples. It only measures the influence of the sample to the parameters, but this is totally different with the quality of the sample itself. Please give a explanation here why these two metrics are related with each other and some experiments to show this.

---

> ### Author Response · Authors · 2025-05-31
>
> ### **Sharpe Ratio choice**
>
> Our primary motivation for selecting the Sharpe ratio stems from its intuitive trade-off between expected benefit (mean gradient magnitude) and potential downside (variance), which aligns well with the goals of active learning under a limited budget.
>
> Compared to other alternatives like the Sortino ratio, which penalizes only downside deviations, the Sharpe ratio has the advantage of being analytically simpler and symmetric in how it treats possible preference outcomes. Accounting for uncertainty of outcomes provides a general-purpose choice. Furthermore, the Sharpe ratio admits a closed-form derivation in our setup (Eq. 15), which enables efficient computation without requiring multiple backpropagations, making it attractive for large-scale training.
>
> That said, we acknowledge that alternative risk-aware metrics such as the Sortino ratio could offer interesting inductive biases. Investigating the impact of such metrics is a promising direction for future work.
>
> ### **Experiments**
>
> Thank you for the observation. While our experiments focus on two benchmarks, we evaluate across three models of varying scale, resulting in six distinct experiments that provide a degree of diversity. We believe this level of evaluation is comparable to or exceeds what is typically reported in prior work on active learning for DPO [1,5,6,7]. Nonetheless, we agree that broader validation would further strengthen the paper. As part of the revision, we will incorporate additional baselines and an ablation study..
>
> ### **Oracle choice**
>
> The use of strong models such as GPT-4o is aligned with current best practices in the alignment literature, primarily because human evaluation at this scale is generally infeasible [1,2,3,4].
>
> Moreover, prior work has shown that strong models like GPT-4o correlate well with human judgments on preference data, making them reasonable proxies for large-scale evaluation. To further mitigate positional bias, we evaluated each response pair in both forward and reversed order and reported the average win rate, a common technique to reduce such artifacts.
>
> We also empirically validated the performance of several available API models on the HH and SHP datasets and selected GPT-4o as the most accurate evaluator available at the time.
>
> [1]Rafailov, Rafael, et al. "Direct preference optimization: Your language model is secretly a reward model." Advances in Neural Information Processing Systems 36 (2023): 53728-53741.
>
> [2 ]Han, Jiaqi, et al. "$ f $-PO: Generalizing Preference Optimization with $ f $-divergence Minimization." AISTATS (2025)
>
> [3] Dubois, Yann, et al. "Alpacafarm: A simulation framework for methods that learn from human feedback." Neurips (2023).
>
> [4] Kirsch, Andreas, and Yarin Gal. "Unifying approaches in active learning and active sampling via fisher information and information-theoretic quantities." TMLR (2022).
>
> [5] William Muldrew, Peter Hayes, Mingtian Zhang, and David Barber. Active preference learning for large language models. ICML, 2024
>
> [6] Ji, Kaixuan, Jiafan He, and Quanquan Gu. Reinforcement learning from human feedback with active queries. corr. (2024).
>
> [7] Xiong, Wei, et al. "Iterative preference learning from human feedback: Bridging theory and practice for rlhf under kl-constraint." ICML (2023).
>
> ### **Gradient Impact**
>
> Thank you for raising this important point. We agree that high gradient magnitude alone does not always correlate with beneficial influence on the model and that relying solely on this metric can introduce undesirable effects such as gradient spikes from noisy or uninformative samples.
>
> Our motivation for considering gradient magnitude stems from its connection to the Fisher Information (FI), which is often used in active learning and uncertainty estimation [4]. Under standard regularity conditions, the FI is the expected squared gradient. Additionally, the inverse of the FI is the Cramer-Rao lower bound on the variance of an unbiased estimator, providing a principled signal for identifying parameters where additional supervision could yield the most significant learning impact. In this sense, gradient magnitude serves as a proxy for identifying high-leverage updates.
>
> However, we recognize that gradient magnitude alone is insufficient. To address this, our method incorporates the Sharpe ratio, which balances the expected gradient magnitude with its variability across preference outcomes. This design mitigates issues associated with gradient spikes. For instance, if one preference leads to a large gradient while the opposite yields a much smaller update, the resulting variance will be high, and the Sharpe ratio will be penalized accordingly. In such cases, the sample is deprioritized despite its large individual gradient, offering a safeguard against misleading updates.
>
> The consistent performance of the proposed approach suggests that using the Sharpe ratio of the gradient as a selection metric has practical merit.

---

> > ### Comment · Reviewer_C2ud · 2025-06-08
> >
> > Thanks for your kind response, I have improved my score. But I still have concern about the experiments. Please show more results on your method.

---

> > > ### Author Response · Authors · 2025-06-10
> > > **Additional Results**
> > >
> > > Thank you for your constructive feedback. As suggested during the review process, we conducted additional experiments, adding a new baseline and an ablation study. The figures are provided in the following link https://anonymous.4open.science/r/SHARP-E983/Sharpe_rebuttal.pdf:
> > >
> > > * We provide comparison results against the APL baseline proposed (Muldrew et al, 2024) for all our experiments. Our method outperforms the baseline in all experiments.
> > >
> > > * We additionally provide a comparison against the expected gradient on the GPT-2 model using both datasets and observe that the SHARP method performs better. Notably, we expect the performance gap to be even larger in noisy datasets that include response pairs where one option is unlikely or nonsensical. Due to the high memory and computational demands of extracting individual gradients for each data point, we were only able to run these experiments with the GPT-2 model within the timeframe of the rebuttal. This computational challenge further underscores the strength of the SHARP approach, as its closed-form expression avoids the need for expensive per-sample gradient computations.
> > >
> > > We sincerely appreciate the valuable suggestions from the review process and believe that these additions meaningfully strengthen our submission. If our rebuttal and the new results address your concerns, we would be grateful if you would consider raising your score.

---

### Official Review · Reviewer_MhkP · 2025-05-11

**Rating:** 6
**Confidence:** 4
**Ethics Flag:** 1

**Summary:**

This paper introduces a novel active learning approach called SHARP aiming at selecting data points for annotation by using a risk assessment strategy based on Sharpe ratio. Drawing inspiration from Sharpe ratio, this method models selection criterion with both the expectation and the variance of the magnitude of gradient update. The paper claims the method could achieve more effective DPO training under same constrained annotation budgets. The paper proves the claim by comparing win rates of DPOs using different data selection methods (SHARP, weighted variant W-SHARP and standard) on HH-RLHF and SHP dataset.

**Questions To Authors:**

Please refer to the reasons to reject.

**Reasons To Accept:**

1.The method provides a new perspective to apply AL to RLHF methods without heuristic preference assumptions.
2.The method outperforms DPO under standard data selection strategy and has a data assessment method that is computationally efficient.

**Reasons To Reject:**

1.The proposed data selection strategy promotes the sample with higher gradient magnitude, which is not necessarily corelated to better influence. Lacks further discussion.
2.Some details in the experiments remain unclear.
a)	What is the data selection strategy for the standard DPO?
b)	There seems to be an inconsistency regarding the definition of win rates. On page 9, win rates are presented as “the winrates against the dataset’s designated chosen completions.” However, the subsequent paragraph indicates that they “compare each newly generated response with the baseline DPO generation.”

---

> ### Author Response · Authors · 2025-05-31
> **Rebuttal**
>
> ### **Gradient Impact**
>
> Thank you for raising this important point. We agree that high gradient magnitude alone does not always correlate with beneficial influence on the model and that relying solely on this metric can introduce undesirable effects such as gradient spikes from noisy or uninformative samples.
>
> Our motivation for considering gradient magnitude stems from its connection to the Fisher Information (FI), which is often used in active learning and uncertainty estimation [1]. Under standard regularity conditions, the Fisher Information (FI) is the expected squared gradient. Additionally, the inverse of the FI is the Cramer-Rao lower bound on the variance of an unbiased estimator, providing a principled signal for identifying parameters where additional supervision could yield the most significant learning impact. In this sense, gradient magnitude serves as a proxy for identifying high-leverage updates.
>
> However, we recognize that gradient magnitude alone is insufficient. To address this, our method incorporates the Sharpe ratio, which balances the expected gradient magnitude with its variability across preference outcomes. This design mitigates issues associated with gradient spikes. For instance, if one preference leads to a large gradient while the opposite yields a much smaller update, the resulting variance will be high, and the Sharpe ratio will be penalized accordingly. In such cases, the sample is deprioritized despite its large individual gradient, offering a safeguard against misleading updates.
>
> The consistent performance of the proposed approach suggests that using the Sharpe ratio of the gradient as a selection metric has practical merit.
>
> [1] Kirsch, Andreas, and Yarin Gal. "Unifying approaches in active learning and active sampling via Fisher information and information-theoretic quantities." TMLR (2022).
>
> ### **Other details**
>
> 2.a. The standard DPO uses a random selection from the dataset. We will make this clear in the paper
>
> 2.b. This is a typo. The correct definition is the first one, "the winrates against the dataset’s designated chosen completions". We will make sure to fix this.

---

> > ### Comment · Reviewer_MhkP · 2025-06-08
> >
> > Thanks for your kind response. I acknowledge the connection between gradient magnitude and the trace of Fisher information. However, given that:
> >
> > 1. The gradient magnitude is a non-negative measure, which can not be a good negative influence indicator.
> > 2. The trace of single-sample empirical Fisher information oversimplifies Fisher information.
> >
> > I believe it is an insufficient explanation. I'll keep my score.

---

> > > ### Author Response · Authors · 2025-06-10
> > > **Additional Results**
> > >
> > > Thank you for your constructive feedback. As suggested during the review process, we conducted additional experiments, adding a new baseline and an ablation study. The figures are provided in the following link https://anonymous.4open.science/r/SHARP-E983/Sharpe_rebuttal.pdf:
> > >
> > > * We provide comparison results against the APL baseline proposed (Muldrew et al, 2024) for all our experiments. Our method outperforms the baseline in all experiments.
> > >
> > > * We additionally provide a comparison against the expected gradient on the GPT-2 model using both datasets and observe that the SHARP method performs better. Notably, we expect the performance gap to be even larger in noisy datasets that include response pairs where one option is unlikely or nonsensical. Due to the high memory and computational demands of extracting individual gradients for each data point, we were only able to run these experiments with the GPT-2 model within the timeframe of the rebuttal. This computational challenge further underscores the strength of the SHARP approach, as its closed-form expression avoids the need for expensive per-sample gradient computations.
> > >
> > > We sincerely appreciate the valuable suggestions from the review process and believe that these additions meaningfully strengthen our submission. If our rebuttal and the new results address your concerns, we would be grateful if you would consider raising your score.

---

### Official Review · Reviewer_MNhU · 2025-05-12

**Rating:** 6
**Confidence:** 2
**Ethics Flag:** 1

**Summary:**

This paper introduces a novel active learning approach for Reinforcement Learning from Human Feedback (RLHF), specifically focusing on improving the efficiency of preference data collection for Direct Preference Optimization (DPO). The authors propose SHARP (SHarpe Ratio-based Active Requested Preferences), a method that uses the Sharpe ratio from statistical finance to evaluate and select the most informative prompt-response pairs for human annotation. The key innovation is assessing the risk-reward tradeoff of potential gradient updates before knowing which response will be preferred. The authors derive closed-form expressions for computing Sharpe ratios efficiently and introduce two variants: SHARP (assuming equal likelihood of preferences) and W-SHARP (incorporating a prior from an implicit reward model). Experiments across multiple language models and datasets demonstrate that their method outperforms standard DPO by up to 5% in win rates against chosen completions while using significantly less labeled data (less than 18% of available training tuples).

**Questions To Authors:**

1) Could you provide additional ablation studies showing how different formulations of the risk component in the Sharpe ratio affect performance? For example, what happens if you use only the expected gradient magnitude without considering variance?

2) Have you conducted experiments applying your method to other RLHF with explicit reward models? If so, how does the performance compare to the results presented in the paper?

3) Have you investigated whether the data selected by your method exhibits any systematic biases compared to randomly selected data? If so, how might these biases affect the model's performance on diverse real-world queries?

4) How does the performance gap between your method and standard DPO change as the labeling budget increases? Is there a point at which the benefits of active learning diminish?

**Reasons To Accept:**

1) The paper introduces a novel and well-motivated approach to active learning for RLHF that alleviates a practical challenge: the high cost of collecting human preference data. The use of the Sharpe ratio as a risk assessment metric for data selection is an innovative application of a concept from finance to machine learning.

2) The authors provide a mathematically rigorous derivation of closed-form expressions for their acquisition functions, making the approach computationally efficient and practical for real-world applications. This avoids the need for multiple backpropagations, which would otherwise make the method prohibitively expensive.

3) The empirical evaluation is comprehensive, testing the approach across different model sizes and on multiple established datasets, demonstrating consistent performance improvements over standard DPO with limited labeled data.

4) The method is model-agnostic and can be integrated into existing RLHF/DPO pipelines without significant modifications, increasing its practical utility for the broader AI community.

**Reasons To Reject:**

1) The paper lacks a thorough ablation study to understand the contribution of different components of the proposed method. For example, it would be valuable to see how performance varies with different definitions of risk or alternative formulations of the Sharpe ratio.

2) While the authors mention that their approach can be applied to traditional RLHF beyond DPO, they only provide experimental results for DPO. This limits the understanding of how well the method generalizes across different RLHF frameworks.

3) The evaluation methodology relies heavily on GPT-4o as an evaluation oracle, which introduces potential biases and limitations. The paper would benefit from additional human evaluation or alternative automated metrics to validate the findings.

4) The paper does not sufficiently address potential distribution shifts that might occur due to selective sampling based on Sharpe ratios. It's unclear how this sampling strategy might bias the model training over time and whether this could lead to performance degradation in certain domains or tasks.

5) The experimental setup is limited to a relatively small training budget, and it's unclear how the benefits of the proposed method scale with larger datasets or longer training regimes.

---

> ### Author Response · Authors · 2025-05-31
> **Rebuttal**
>
> ### **Expected Gradient magnitude**
>
>  Thank you for suggesting this ablation study. Although we have run a few hand-picked examples to validate our motivation about the usefulness of the Sharpe ratio over the expected gradient magnitude, we did not run a full comparison on our experiments. We are currently running this ablation and will include the results in the updated version of the paper or the rebuttal if available in time.
>
> ### **Explicit Reward Model**
>
> Thank you for raising this point. We did not conduct experiments using an explicit reward model as part of the RLHF pipeline. Our motivation is twofold:
>
> First, applying active learning to explicit reward models effectively reduces the problem to traditional active learning aimed at improving the reward model itself, rather than directly optimizing the LLM policy. This departs from our goal, which is to guide the acquisition process based on the policy update impact rather than intermediate components like the reward model.
>
> Second, a central contribution of our work is the derivation of a closed-form expression for the Sharpe ratio in the DPO setting, which enables efficient computation in terms of both runtime and memory. While it may be possible to extend the Sharpe ratio framework to other variants of RLHF or DPO that rely on explicit reward models, doing so efficiently without facing significant memory constraints (due to individual gradient extraction) is nontrivial and would represent additional contributions. We consider this a valuable direction for future work, but outside the scope of the current paper.
>
> ### **Selection Bias**
>
> Thank you for raising this important question. In our current study, we did not perform a formal analysis of systematic biases in the data selected by our method compared to random sampling. Our focus was primarily on evaluating the effectiveness of Sharpe ratio-guided selection in improving policy performance under constrained labeling budgets. That said, we acknowledge that active learning methods, by design, introduce a selection bias. While this can be beneficial for training efficiency, it may also reduce coverage of more typical or diverse user queries. As noted in our discussion of limitations, our method may shift the distribution of selected data, and future work could explore techniques such as importance weighting or distributional regularization to mitigate potential coverage gaps.
>
> ### **Active Learning Budget**
>
> Thank you for the insightful question. In our current experiments, we focused on the low-data regime, where efficient use of the labeling budget is most critical. In this setting, both SHARP and W-SHARP consistently outperform standard DPO.
>
> While we have not conducted a full sweep across budget sizes, we expect the marginal benefit of data selection to diminish as more labeled data becomes available. This is a well-known characteristic of active learning, where its advantages are most pronounced under restricted labeling budgets. In the high-budget regime, random sampling may become increasingly competitive as the policy already receives diverse and sufficient supervision.

---

> > ### Comment · Reviewer_MNhU · 2025-06-09
> >
> > Thank you for the authors' response. Some of my concerns have been addressed. I'll keep my positive score.

---

> > > ### Author Response · Authors · 2025-06-10
> > > **Additional Results**
> > >
> > > Thank you for your constructive feedback. As suggested during the review process, we conducted additional experiments, adding a new baseline and an ablation study. The figures are provided in the following link https://anonymous.4open.science/r/SHARP-E983/Sharpe_rebuttal.pdf:
> > >
> > > * We provide comparison results against the APL baseline proposed (Muldrew et al, 2024) for all our experiments. Our method outperforms the baseline in all experiments.
> > >
> > > * We additionally provide a comparison against the expected gradient on the GPT-2 model using both datasets and observe that the SHARP method performs better. Notably, we expect the performance gap to be even larger in noisy datasets that include response pairs where one option is unlikely or nonsensical. Due to the high memory and computational demands of extracting individual gradients for each data point, we were only able to run these experiments with the GPT-2 model within the timeframe of the rebuttal. This computational challenge further underscores the strength of the SHARP approach, as its closed-form expression avoids the need for expensive per-sample gradient computations.
> > >
> > > We sincerely appreciate the valuable suggestions from the review process and believe that these additions meaningfully strengthen our submission. If our rebuttal and the new results address your concerns, we would be grateful if you would consider raising your score.

---

### Official Review · Reviewer_ZGMy · 2025-05-14

**Rating:** 6
**Confidence:** 4
**Ethics Flag:** 1

**Summary:**

This paper presents an active learning method to acquire labeled preference tuples for LLM fine-tuning.

For each unlabeled preference tuple (consisting of a prompt and two candidate responses), they compute the magnitude of the the DPO gradient update under the two possible preference orderings, then they compute the Sharpe ratio (average / standard deviation) of these two values, and use this as an acquisition function to submit the tuple to the human annotator for labeling. The authors also provide a derivation which enables them to compute the Sharpe ratio efficiently by only doing one DPO computation rather than two.

They evaluate their methods on standard LLMs and standard datasets, comparing the win rate of the preferred response on a test set over vs the number of training examples, comparing against DPO (presumably with randomly selected examples) as the baseline, showing improvements.

While the method is interesting, and, as far as I know, novel, I notice a few areas of concerns:
- The authors mention running SFT on the positive examples before DPO or their methods, as it is standard practice, but how are the SFT examples labeled? If they use all the dataset, then this defeats the purpose of doing active learning.
- The improvements are very small, and the authors only consider DPO, with presumably random chosen examples, as the baseline. How does their method compare to other active learning approaches?
- I'm not sure I buy the argument for using the Sharpe ratio as opposed to the unnormalized magnitude of the gradient updates. This has to be evaluated empirically, and the authors did not do it.

**Questions To Authors:**

- How are the SFT examples labeled?
- How does using the expected gradient magnitude as opposed to the Sharpe ratio perform?

**Reasons To Accept:**

Interesting, novel, improvements.

**Reasons To Reject:**

Weak baseline, small improvements, no relevant ablations.

EDIT:

The authors reported more baselines and ablations.

---

> ### Author Response · Authors · 2025-05-31
> **Rebuttal**
>
> ### **SFT Labels**
> Thank you for the comment. In our evaluation, we use synthetic datasets that provide a prompt, a preferred answer, and one or more less-preferred alternatives. Following standard practice in the alignment literature [1,2,3,4], we treat the preferred answer as the SFT label. To simulate a realistic active learning scenario, where the prompt–response pairs used for collecting preferences differ from those seen during the SFT phase, we partition the dataset accordingly: a small fraction is used for SFT, and the remainder is reserved for DPO and active learning. Our setup is intended to reflect the practical setting where one begins with a labeled dataset and subsequently searches for additional informative pairs to label.  We will update the paper to clarify this distinction.
>
> [1]Rafailov, Rafael, et al. "Direct preference optimization: Your language model is secretly a reward model." Advances in Neural Information Processing Systems 36 (2023): 53728-53741.
>
> [2 ]Han, Jiaqi, et al. "$ f $-PO: Generalizing Preference Optimization with $ f $-divergence Minimization." AISTATS (2025)
>
> [3] Dubois, Yann, et al. "Alpacafarm: A simulation framework for methods that learn from human feedback." Advances in Neural Information Processing Systems 36 (2023): 30039-30069.
>
> [4]Meng, Yu, Mengzhou Xia, and Danqi Chen. "Simpo: Simple preference optimization with a reference-free reward." Advances in Neural Information Processing Systems 37 (2024): 124198-124235.
>
> ### **Experiments**
> Thank you for this suggestion. Due to the cost associated with using the evaluation API, we initially focused our evaluation on the most relevant baseline to highlight the core contributions of our method.
>
> We agree that including additional baselines would strengthen the empirical analysis. To address this, we are currently implementing an additional baseline within our codebase to ensure a fair and consistent comparison. We will share the corresponding results as soon as they are available and will include them in the rebuttal if they are ready in time.
>
> ### **Expected Gradient magnitude**
>
> Thank you for suggesting this ablation study. Though we have run a few hand-picked examples to validate our motivation about the usefulness of the Sharpe ratio over the expected gradient magnitude, we did not run a full comparison on our experiments. We are currently running this ablation and will include the results in the updated version of the paper or the rebuttal if available in time.

---

> > ### Comment · Reviewer_ZGMy · 2025-06-07
> >
> > Thanks for your response.
> >
> > If you don't have any more experimental results, I'll keep my score.

---

> > > ### Author Response · Authors · 2025-06-10
> > > **Additional Results**
> > >
> > > Thank you for your constructive feedback. As suggested during the review process, we conducted additional experiments, adding a new baseline and an ablation study. The figures are provided in the following link https://anonymous.4open.science/r/SHARP-E983/Sharpe_rebuttal.pdf:
> > >
> > > * We provide comparison results against the APL baseline proposed (Muldrew et al, 2024) for all our experiments. Our method outperforms the baseline in all experiments.
> > >
> > > * We additionally provide a comparison against the expected gradient on the GPT-2 model using both datasets and observe that the SHARP method performs better. Notably, we expect the performance gap to be even larger in noisy datasets that include response pairs where one option is unlikely or nonsensical. Due to the high memory and computational demands of extracting individual gradients for each data point, we were only able to run these experiments with the GPT-2 model within the timeframe of the rebuttal. This computational challenge further underscores the strength of the SHARP approach, as its closed-form expression avoids the need for expensive per-sample gradient computations.
> > >
> > > We sincerely appreciate the valuable suggestions from the review process and believe that these additions meaningfully strengthen our submission. If our rebuttal and the new results address your concerns, we would be grateful if you would consider raising your score.

---

> > > > ### Comment · Reviewer_ZGMy · 2025-06-10
> > > >
> > > > Thanks.
> > > >
> > > > I have increased my score.

---

### Decision · Program_Chairs · 2025-07-08

**Decision:**

Accept

**Comment:**

This paper introduces a new way to do RLHF, based on an idea from finance (Sharpe ratio) to select from the preference data. The authors use a closed-form solution for this, and it runs efficiently. The new ablation study and new baseline demonstrate the advantage of the approach.

This paper seems to be in possibly a good shape for publication, even though most of the reviewers asked for additional experiments. The authors added a specific set of experiments. I am not completely certain it addresses all the requests (it is a limited set of experiments that tries to address all of the requests), but reviewers, overall, seem to be satisfied.